# Daily Accessed Street Greenery and Housing Price: Measuring Economic Performance of Human-Scale Streetscapes via New Urban Data

**Yu Ye [1],\*** , **Hanting Xie [2]** , **Jia Fang [1],\*** , **Hetao Jiang [1]** and **De Wang [1]**

1   College of Architecture and Urban Planning, Tongji University, Shanghai 200092, China;
    jiang_hetao@foxmail.com (H.J.); dewang@tongji.edu.cn (D.W.)
2   School of Design, University of Pennsylvania, Philadelphia, PA 19104, USA; hantingx@upenn.edu
\*   Correspondence: yye@tongji.edu.cn (Y.Y.); 07fangjia@mail.tongji.edu.cn (J.F.)

**Abstract:** The protective effects of street greenery on ecological, psychological, and behavioral phenomena have been well recognized. Nevertheless, the potential economic effect of daily accessed street greenery, i.e., a human-scale and perceptual-oriented quality focusing on exposure to street greenery in people's daily lives, has not been fully studied because a quantitative measuring of this human-scale indicator is hard to achieve. This study was an attempt in this direction with the help of new urban data and new analytical tools. Shanghai, which has a mature real estate market, was selected for study, and the housing prices of 1395 private neighborhoods in its city center were collected. We selected more than forty variables that were classified under five categories—location features, distances to the closest facilities, density of facilities within a certain radius, housing and neighborhood features, and daily accessed street greenery—in a hedonic pricing model. The distance and density of facilities were computed through a massive number of points-of-interest and a geographical information system. The visible street greenery was collected from Baidu street view images and then measured via a machine-learning algorithm, while accessibility was measured through space syntax. In addition to the well-recognized effects previously discovered, the results show that visible street greenery and street accessibility at global scale hold significant positive coefficients for housing prices. Visible street greenery even obtains the second-highest regression coefficient in the model. Moreover, the combined assessment, the co-presence of local-scale accessibility and eye-level greenery, is significant for housing price as well. This study provides a scientific and quantitative support for the significance of human-scale street greenery, making it an important issue in urban greening policy for urban planners and decision makers.

**Keywords:** daily accessed street greenery; housing price; street view images; human scale; new urban data; Shanghai

## 1. Introduction

### 1.1. The Importance of Daily Accessed Street Greenery and the Difficulty of Measuring It

As an important component of urban green infrastructure, street greenery, i.e., street trees, shrubs, lawns, and other forms of vegetation along streets, has long been recognized for its multiple benefits for urban residents. Street greenery's ecological benefits have been well recognized, including its role in producing oxygen [1], mitigating smog [2], controlling urban heat island effects [3], and reducing noise pollution [4]. Recently, the psychological and behavioral benefits of street greenery have also been found. The perception of street trees contributes to positive psychological effects, e.g., relaxation

and pleasure [5], as well as the esthetic rating of urban places [6]. It also significantly contributes to increasing the walkability of urban streets and improving social behaviors there [7,8].

Nevertheless, further exploration of the economic performance of daily accessed street greenery is constrained to certain extent because an objective measurement of this issue is hard to achieve. The definition of daily accessed street greenery contains both visual and accessible dimensions [9], which can be regarded as a series of indicators including not only purely visible greenery and street accessibility but also combined categories based on the high or low values of both greenery and accessibility. In short, it is a human-scale, perceptual-oriented quality focusing on exposure to street greenery in people's daily lives. Specifically, visible street greenery has been hard to determine because it is a form of data coming from a human viewpoint. Some pioneering studies have been made using color photographs and greenery extraction, but these take time and are inefficient [10,11]. Taking photos and calculating the percentage of green pixels can only be used at the neighborhood scale, which would not support a large-scale, quantitative mapping of visible street greenery. The issue of accessibility, meanwhile, has been well discussed for decades in network theory and transportation studies [12]. Nevertheless, the calculation of accessibility was usually based on the shortest physical path or travel time while ignoring human preference [13]. To date, the geographical accessibility of green spaces, such as parks and woodlands, has been well studied through the application of geographic information system [14]. However, less research has focused on the accessibility of roadside vegetation, which is a highly visible form of urban vegetation that many residents may pass through and experience daily [15].

## 1.2. Hedonic Price Model and Its Recent Developments

As one of the most commonly applied methods for housing price evaluation, the hedonic price model was put forward by Rosen in the 1970s [16]. This model argues that an item can be valued by its internal characteristics and that each characteristic will carry a unique implicit price in an equilibrium market. Therefore, the total price of an item can be estimated as the aggregated prices of its internal attributes or services, such as its structure, neighborhood, and environmental characteristics [17]. On this basis, the value of a house can be divided into several constituent parts, and the significance or value of each part can be measured according to the correlation coefficients.

Following this approach, many key attributes of the built environment have been included in the study of housing prices decades ago, including geographical locations, facility conditions, and housing and neighborhood characteristics [18–20]. In recent years, the availability of built environment data allows researchers to explore environmental externalities affecting housing price through the hedonic price model. The value of urban wetland [21], scenic views [22], air quality [23], public safety [24], and the distance to many facilities including urban parks and bus stops [25,26], have been studies in many cities over the world. Besides the extension of research focuses, new regression models such as hierarchical linear model (HLM) have been used in the study of housing price, which show higher goodness of fit compared with traditional models such as ordinary least squares (OLS) [27].

Moreover, some recent empirical studies have shown that it is inappropriate to equate distances to facilities with accessibility. Consequently, some researchers have introduced the measurement of street network accessibility by space syntax tools, e.g., sDNA, to establish multi-dimensional hedonic price models [28,29]. In contrast to traditional models focusing on the geometrical or compositional attributes, those including street accessibility aim at measuring the urban configuration in terms of actual functions, which may improve the reliability of the analytical models. In addition, since the street-based block was verified as having a more significant effect than the region-based or grid-based units when analyzing local house prices [30], it might also be worth considering the street-based blocks in order to examine local situations more precisely.

In short, this trend of adding fine-scale attributes into the study of housing prices performs well. Nevertheless, the attribute belonging to the human-scale streetscape has been neglected in previous studies. Even so, a few related studies have been made at either the neighborhood scale [31]

or on the relationship between eye-level greenery and housing prices, but they have ignored the issue of accessibility in people's daily lives [32]. A better understanding of this area, with a broader perspective, would provide important supplement in urban greening policy for urban planners, developers, and decision makers.

*1.3. New Research Potentials in the Context of New Urban Data and New Tools*

The recent emergence of new urban data, e.g., large amounts of geo-referenced data provided by street view images and open street maps, and of new analytical tools, e.g., machine-learning algorithms and space syntax tools, bring new research possibilities [33,34].

On hte one hand, the integration of street view images and appropriate machine-learning algorithms permits a new approach to the human-scale measuring of spatial features on streets. The availability of large datasets of panoramic images taken at street level, such as Google street view images or Baidu street view images, has provided the data foundation for analyzing human-scale spatial features on streets [35]. Based on that, street view images have been used to measure street greenery via color bands [36]. Subsequently, the rise in machine-learning algorithms has brought computer vision tools to extract visible greenery and other spatial features on streets [37]. For instance, SegNet, a deep convolutional network for achieving an objective and accurate pixel-width image segmentation, might help in the accurate extraction of greenery from street view images, reflecting a complex built environment [38]. Moreover, studies assisted by machine-learning algorithms are going further in measuring that which was considered "unmeasurable" in the past. Many intangible, perceptual qualities on streets, such as visual quality [39], façade maintenance, and the continuity of street walls [40], walkability affected by visible greenery [41,42] have been achieved.

On the other hand, space syntax theory and the availability of analytical tools, such as Depthmap, produced by UCL, and sDNA, produced by Cardiff University, can be used to estimate accessibility and predict movement flows within a given street network configuration [43]. The measuring of street accessibility is achieved through space syntax analysis. Space syntax is a combination of theories on spatial configuration and tools assisting quantitative analysis of street accessibility. A series of distance metrics—metric distance (measuring paths with the shortest length), topological distance (measuring paths with the fewest turns), and geometric distance (measuring paths with the least angular change)—have been developed to represent accessibility [44]. Empirical studies over the last decade have shown that geometric analytics with a metric radius work best in representing "through-movement" potentials, i.e., the potential of each segment element to be selected by pedestrians or drivers as their path [45,46]. This measurement, called "choice" in Depthmap and "betweenness" in sDNA, could be used to predict the most easily accessed streets in people's daily lives.

In short, new research potential is emerging in the context of new data and new analytical tools for exploring the economic influence of daily accessed visible street greenery. Although a small number of studies have analyzed the relationship between eye-level visual greenery and housing price [31,32], a human-centered combination of accessibility in daily lives and visible street greenery is still unexamined. It is also interesting to compare this newly proposed, human-scale indicator with other key attributes of the built environment, such as distance to facilities and housing and neighborhood features, that are usually provided from a top-down viewpoint. A better understanding of this area may assist in more efficient and more human-oriented policy making in relation to urban greenery.

In light of this, we attempted to measure daily accessed street greenery and to identify its degree of influence on housing price by comparing it with several other traditional indicators in this study. Specifically, we regarded the mean housing prices among different street blocks as a proxy for housing price and introduced a hedonic price model for analysis. Since the model assumes that the total housing price of a selected residential unit can be considered as the aggregate prices of its implicit attributes or services of the adjacent built environment, we assume that the indicators of daily accessed street greenery may have some influence as well.

To achieve an in-depth understanding of the economic performance of daily accessed street greenery, this study: (1) quantified visible street greenery using images from Baidu street view across the case study city of Shanghai; (2) quantified street accessibility with different radii, using space syntax tools; and (3) ran a hedonic price model with visible street greenery, street accessibility, combined assessment of the former two, and other classical impact factors to compare their correlation coefficients. This analysis may provide us with a valid demonstration of the relative importance of human-scale indicators on streets.

## 2. Research Methodology

### 2.1. Analytical Framework

To identify the relative economic impact of daily accessed street greenery on the local housing market, this study selected visible street greenery and street accessibility as the major study objects in regression models and introduced a series of conventional indicators as the reference. All the regression coefficients were standardized for comparison.

First, housing price data were collected from the Homelink Real Estate website (Lianjia, the largest real estate website in mainland China). Then, conventional indicators that have been frequently employed in the regression models in previous urban studies were taken, such as housing and neighborhood features, the density of facilities within a certain radius, and distance to the closest facilities. These indicators are capable of interpreting physical settings or urban functions quantitatively, calculated through points-of-interest (PoIs) collected from Baidu Map.

We then introduced the indicators of daily accessed street greenery and visible greenery into the analysis. Such human-scale indicators based on people's daily lives have rarely been considered in previous studies. The measurement of visible greenery was achieved by the combined application of a SegNet algorithm and many Baidu street view images. The measurement of global and local street accessibility was achieved using space syntax. The indicator system is shown in Figure 1.

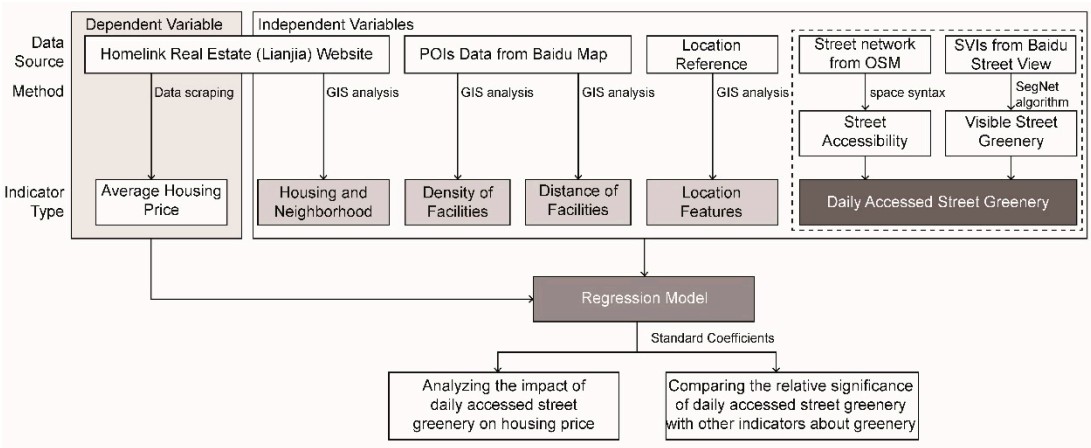

**Figure 1.** Indicator system and analytical framework.

With the help of regression analysis, the study explored the effects between the human-scale, daily accessed street greenery and the housing price, then compared its standardized coefficient with other involved indicators. This helps to answer two questions: (1) To what extent would this kind of human-scale indicator affect housing price? (2) Would this kind of human-scale, perceptual-oriented indicator play a more important role than city-scale indicators related to greenery, such as distance to urban parks and the density of urban parks within a certain radius?

## 2.2. Case Selection

For this research, we selected the Shanghai Middle Ring Road area, of around 400 km$^2$, as the study area (Figure 2). As one of the most well-developed and high-density urban areas in China, this study area has a real estate market that is relatively mature. In other words, the real estate market of this area is supposed to be relatively steady, with less external market disturbance than in other cities. Private housing neighborhoods in street blocks were selected as our analytical units. Other indicators were also transformed to the same study unit for further analysis.

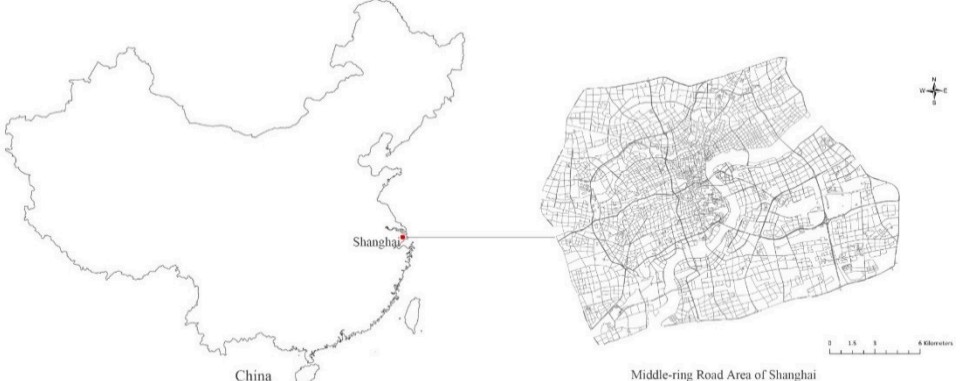

**Figure 2.** Location of Shanghai and the study area: Middle Ring Road area of Shanghai.

## 2.3. Variables and Data

The mean property price of the private neighborhoods was considered as the dependent variable and a series of urban indicators were considered as independent variables. The data sources and quantification methods of the variables are introduced below.

### 2.3.1. Dependent Variable

The dependent variable for this study was the mean housing price for each private neighborhood. The data were crawled from the Homelink Real Estate website (Lianjia) in late 2016 using the Scrapy framework. By projecting the residential units onto the base map according to their geo-references and spatially joining them to their corresponding street blocks, we could obtain property information at the street-block level. In this case, the mean prices for 1395 private neighborhoods in the Shanghai Middle Ring Road area were collected. The average housing prices inside communities and their spatial distribution are shown in the choropleth maps in Figure 3. The price herein was measured as Chinese currency (RMB) per square meter. Moreover, the number of collecting points in Figure 3a is a little bit larger than the number of street blocks in Figure 3b. That was caused by two reasons: (1) some residential communities were built in phases and thus left several records on the website; and (2) several street blocks contained more than one parcels. In those communities containing more than one record, the final record of building age and property price were averaged among these collecting points.

As shown in Figure 3, the price decreases correlate with color changes from dark pink to light pink. The grey color in Figure 3b means a lack of data on housing price. These grey blocks usually obtain other urban functions, such as commercial facilities or green parks. As shown below, the overall distribution of housing property values tends to decrease as distance from the city center increases.

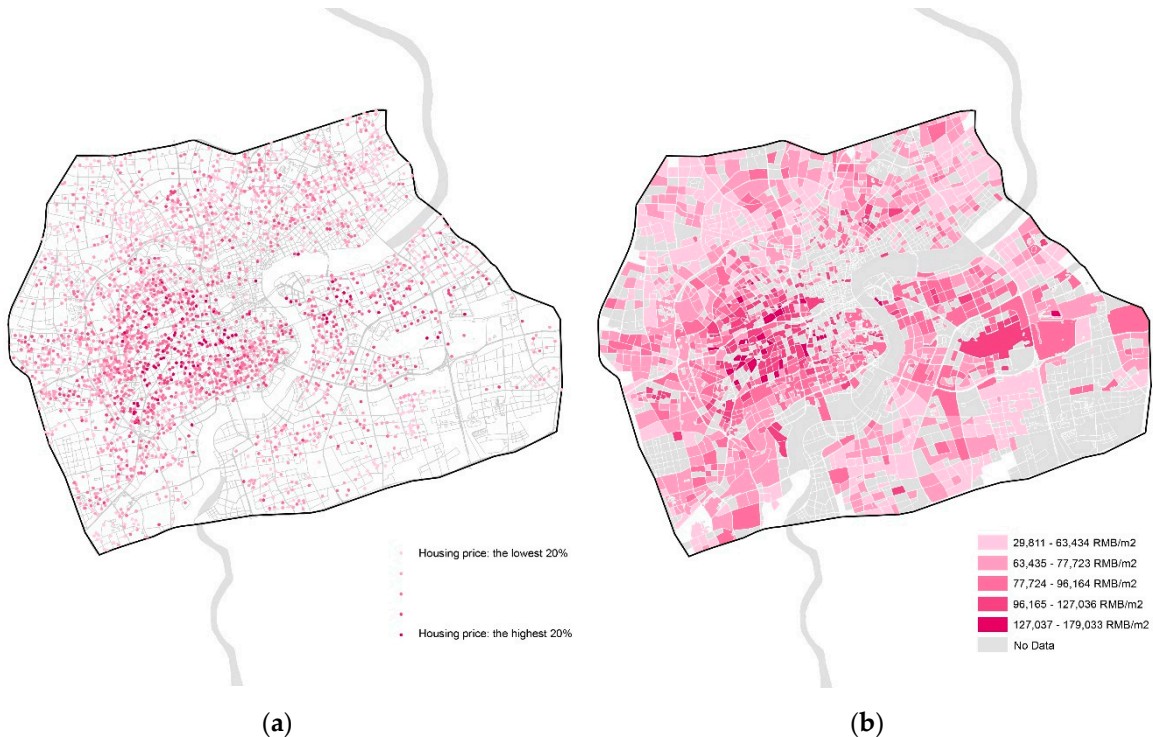

**Figure 3.** Housing prices collected though open data: (**a**) spatial distribution of collecting points of housing price; and (**b**) average housing price for each neighborhood measured as RMB/m². 

### 2.3.2. Independent Variables

Since the hedonic price model theory put forward by Rosen in 1974, many studies have tried to utilize it in practical studies of factors influencing housing prices. One of the earliest dates back to 1979, when Witte and his colleagues attempted to conduct the empirical measurement of Rosen's economic theory by using demographic and locational data [19]. With the development of GIS, a range of types of spatial data have been introduced into hedonic price models for analysis, including density of facilities within a certain radius, distance to closest facilities, and the attributes of facilities nearby [47–49].

In contrast to previous studies, this study extended the hedonic price models by introducing human-scale indicators of daily accessed street greenery, i.e., visible street greenery, street accessibility, and a combination of both (Figure 4). These indicators supplement former models by adding perceptual feelings on streets and extend the hedonic price model to the perspective of eye level and human scale.

- Collecting housing and neighborhood conditions via open data and survey data

The housing and neighborhood conditions were collected from two data sources. The average age of a neighborhood was collected from the Homelink Real Estate website. Then, the average floor, gross building area, building density, and floor area ratio (FAR) were calculated according to the building footprint and floor number provided by local municipality.

- Measuring density and distance of facilities via points-of-interest (PoIs) data and spatial analysis

The density and distance of facilities were calculated through the PoIs data collected from Baidu Map and spatial analysis in GIS. Specifically, the PoIs data (1,815,088 records in total) was separated into 11 categories: school, university or college, middle school, elementary school, kindergarten, bus stop, tourist attraction, park or plaza, shopping mall, hospital, and subway station. The data were used to quantify functional characteristics, including the density of facilities within a certain radius and the distance of each private neighborhood to its closest facilities. Specifically, the radii used in density calculation were selected as 500 m and 1000 m, respectively, according to the urban policy of 15-min community life circle in China [50].

- Measuring location features via open street map (OSM) and spatial analysis

To define the locational factors of each analytical unit, we selected the most important two locational objects in Shanghai, namely, the Huangpu River and the city center, due to their effects on centrality and landscape view. We then quantified the locational factors of each private neighborhood by calculating the distances between the centroid of each private neighborhood and the referenced locational points.

- Measuring daily accessed street greenery towards analytical units

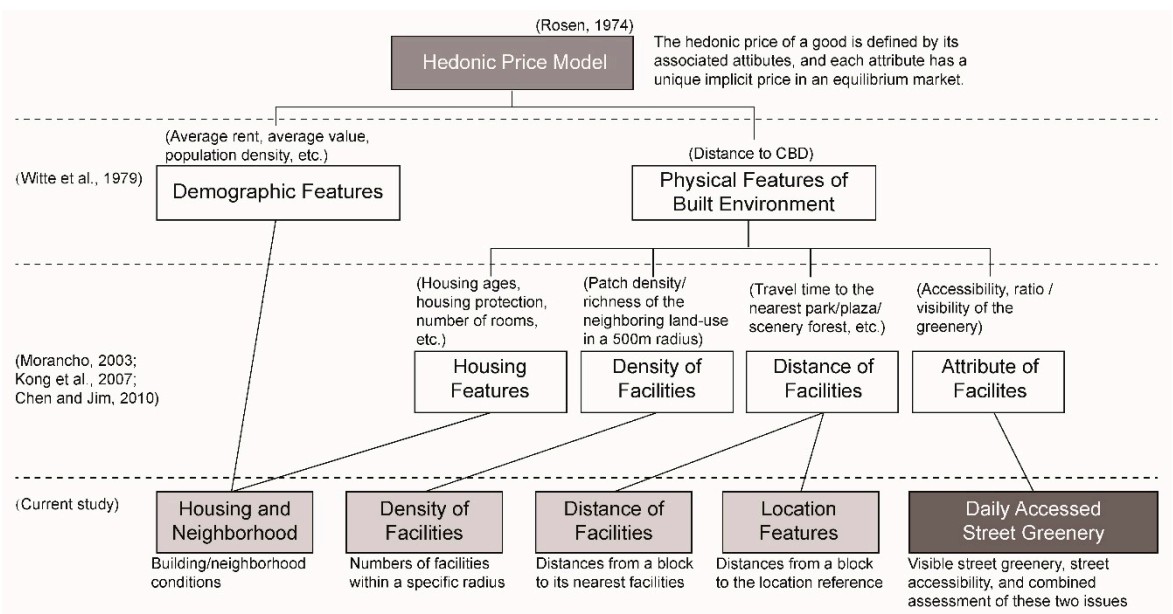

**Figure 4.** The selection of detailed indicators.

(1) Visible street greenery

The measurement of visible street greenery started from the collection of street view images (SVIs) from Baidu, one of the largest internet companies in China providing web services [51]. To achieve a comprehensive representation of spatial features across the site's streetscape, samples from Baidu street view images were collected approximately every 40 m. The 69,137 sample sites were generated along streets from the OSM street network with a total length of 2,611,079 m, in the city center of Shanghai. Each sample site contained four SVIs with the size of $480 \times 360$ pixels were enough to achieve a panoramic view of the surrounding environment (Figure 5).

Specifically, the SVIs were requested in an http URL form using the Baidu Maps API. By defining the URL parameters, users can obtain a static image from any direction and angle of view, for any point where SVIs are available. Within these, we measured street greenery as experienced by people by applying the setting that performed well in Singapore's street greenery study [9]. On the horizontal level, each sample site contained four directional scenes that can act as a proxy of the visible street greenery at a human scale. The images' headings were initially calculated through the topological features computed from street networks and then passed to the Baidu Maps API to ensure that the front and rear views were always parallel with the street and that the left-hand and right-hand views were always vertical with the street. On the vertical level, a zero-degree vertical view angle was downloaded to compute the visually available greenery, in line with related psychological studies [52]. The images applied in our study were crawled during the period of spring 2017. With the help of the time tags on the street view images, we were able to either remove or replace the street view images from the winter to control for the variation in greenery proportions in different seasons.

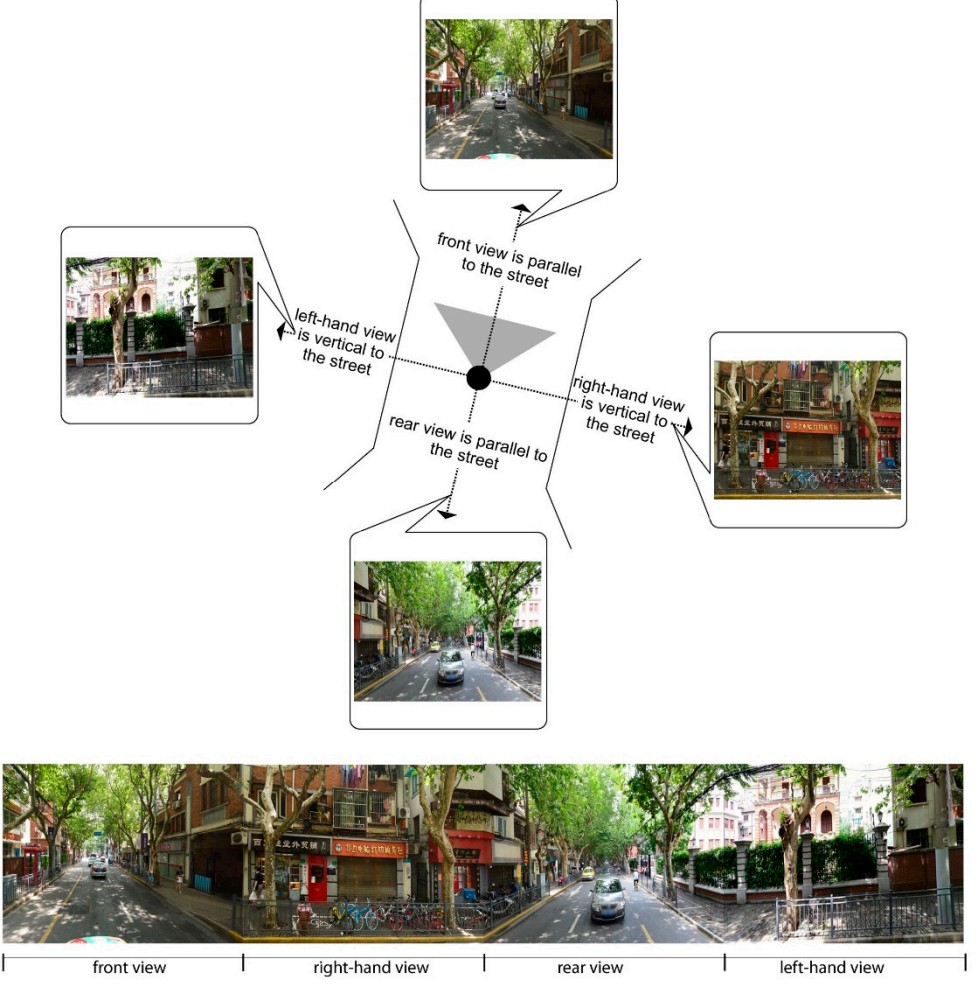

**Figure 5.** The collection of street view images.

After that, we employed the SegNet algorithm to measure the proportions of vegetation elements for each image. Compared to other deep learning algorithms, SegNet is efficient in terms of both memory and computational time. According to its producer, SegNet can achieve a global accuracy of 90.40% can be achieved for a total of 12 classes, and the accuracy is even higher for the classes of building, sky, car, and road [38]. Meanwhile, it is capable of achieving good segmentation performance utilizing low-resolution images, which has been verified by other studies using Chinese cities as the sites [37].

We input all SVIs collected (approximately 230,000 images) into SegNet and interpreted them into colored categories via the SegNet decoder. The vegetation was marked with green and other items were marked with black, as shown in Figure 6. After the calculation of visible greenery on street view images and sample sites, we were then able to calculate the value along a single street segment as an overall greenery index. The transformation from streets, a kind of polyline, into neighborhoods, a kind of polygon, was achieved via a distance decay model. The results are shown in Figure 7.

The equation calculating each block's configuration was

$$G_n = \sum_{i=1}^{n} G_i \frac{L_i D_i^a}{\sum_{i=1}^{n} L_i D_i^a} \tag{1}$$

where $G_n$ is the visible greenery value of each block, $G_i$ is the visible greenery values of the surrounding streets, $L_i$ is the lengths of the street central lines affecting the blocks, $D_i$ is the shortest Euclidian distance from the street central lines to the block edges, and $a$ is the distance decay parameter.

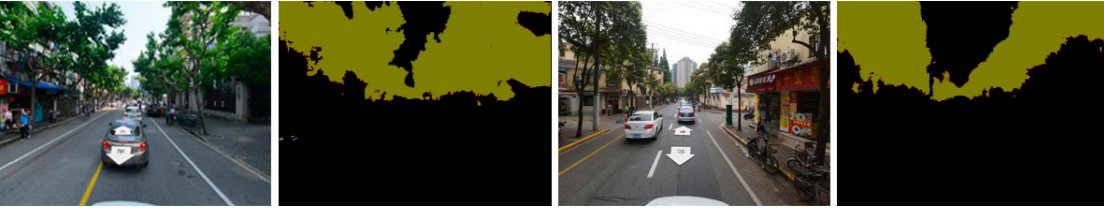

**Figure 6.** The extraction of visible street greenery.

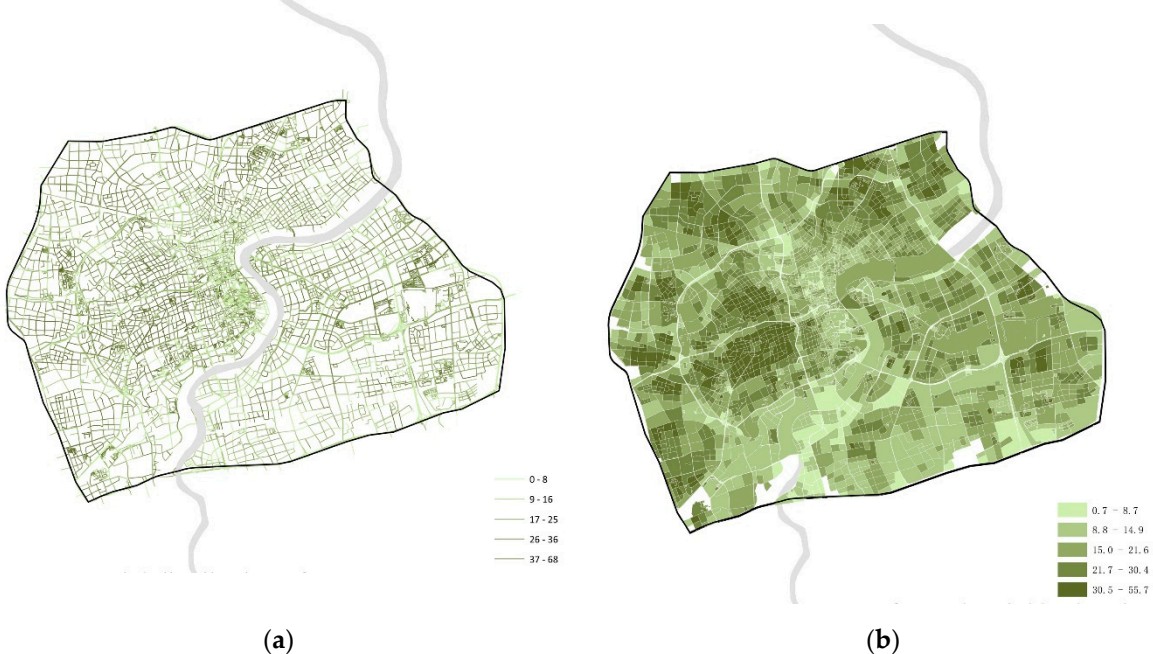

(**a**)          (**b**)

**Figure 7.** The visible greenery on streets and its transformation into neighborhoods: (**a**) visible street greenery; and (**b**) average street greenery for each neighborhood.

(2)    Measuring street accessibility via space syntax

As stated above, the measurement representing "through-movement" potentials, i.e., the potential of each segment element to be selected by pedestrians or drivers as their path, was used to predict the most easily accessed streets in people's daily lives. This measurement is called "choice" in Depthmap and "betweenness" in sDNA.

The choice (betweenness) was defined by Hillier and Iida [53] as:

$$C_b(P_i) = \sum_{j=1}^{n} \sum_{k=1}^{n} g_{jk}(p_i) / g_{jk}(j < k) \tag{2}$$

where $g_{jk}(p_i)$ is the number of geodesics between node $p_j$ and $p_k$ that contain node $p_i$, and $g_{jk}$ is the number of all geodesics between $p_j$ and $p_k$.

More specifically, we used the spatial design network analysis (sDNA) tool to operationalize the computing of the "choice" value. Two radii of accessibility, i.e., 800 m and 6200 m, were used to represent accessibility at local and global scales. Herein, the radius refers to the metric distance from each segment along all of the available streets and roads from that segment up to the radius distance [46]. Thus, a low radius, such as 800 m, mainly identifies the local-scale relationship among streets, i.e., the frequently chosen streets in pedestrian behaviors. A high radius means that a broader area is considered and that main streets in commuting behaviors are highlighted (Figure 8). The selection for calculating radii was made according to traffic surveys in Shanghai [54].

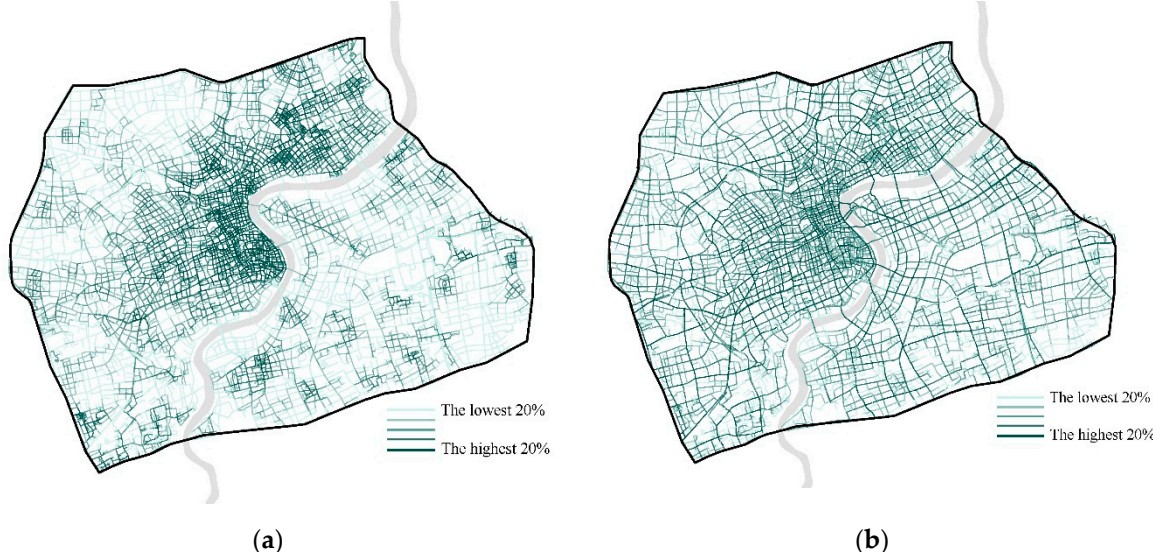

(**a**)            (**b**)

**Figure 8.** Analyzing street accessibility through sDNA: (**a**) street accessibility at local scale: pedestrian-oriented; and (**b**) street accessibility at global scale: commuting-oriented.

(3) The combined assessment of visible street greenery and street accessibility

The combined assessment of visible street greenery and street accessibility was achieved through overlay analysis (Figure 9). Greenery and accessibility at global and local scales were classified as the first and second halves and then combined as different types. Specifically, there were three main types: Type I (high value) represented both high values in visible street greenery and street accessibility; Type II (medium value) represented one high value and one low value in these two dimensions; and Type III (low value) represented both low values. In this way, we were able to produce a categorical classification representing both greenery and accessibility at human scale.

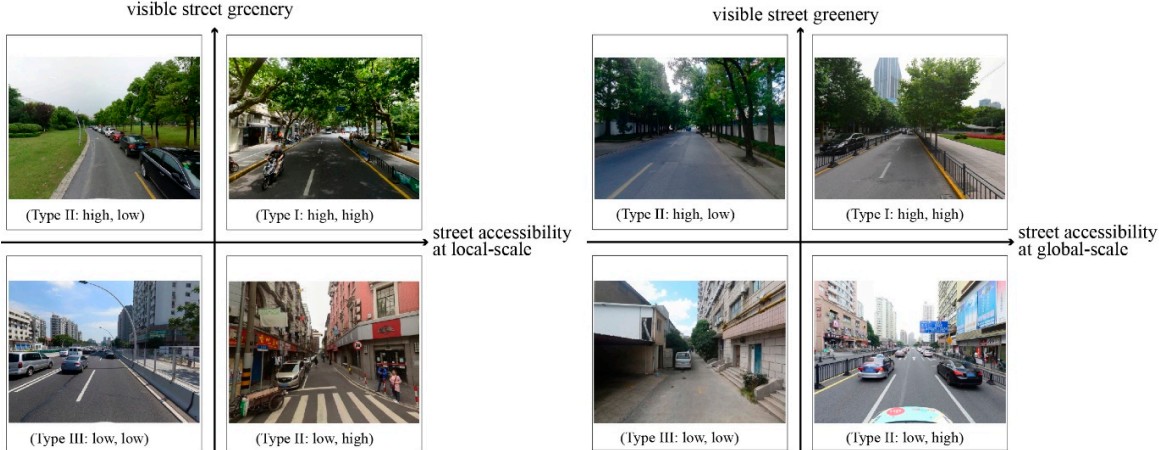

**Figure 9.** The classified types of combined visible street greenery and street accessibility.

The description and spatial distribution of each variable are listed in Table 1.

**Table 1.** Description of the variables.

| Indicator Types | Variables | Description and Distribution |
|---|---|---|
| *Housing and neighborhood features* | Avg_age<br>Avg_floor<br>Sum_bldA<br>Sum_arch_a<br>Arch_den<br>FAR | Average age of the private neighborhood.<br>Average building stories of the private neighborhood.<br>Gross building area of the private neighborhood.<br>Total floor area of the private neighborhood.<br>Building density of the private neighborhood.<br>Floor area ratio of the private neighborhood.<br> |
| *Density of facilities (PoIs)* | School_5H<br>University_5H<br>Middleschool_5H<br>Primaryschool_5H<br>Kindergarten_5H<br>Busstop_5H<br>Attraction_5H<br>Parkplaza_5H<br>Shopping_5H<br>Hospital_5H<br>Subway_5H | The number of PoIs within the 500 m buffer of the neighborhood centroid.<br> |
| | School_1K<br>University_1K<br>Middleschool_1K<br>Primaryschool_1K<br>Kindergarten_1K<br>Busstop_1K<br>Attraction_1K<br>Parkplaza_1K<br>Shopping_1K<br>Hospital_1K<br>Subway_1K | The number of PoIs within the 1000 m buffer of the neighborhood centroid.<br> |

**Table 1.** *Cont.*

| Indicator Types | Variables | Description and Distribution |
| --- | --- | --- |
| *Distance of facilities (PoIs)* | d_School<br>d_University<br>d_Middleschool<br>d_Primaryschool<br>d_Kindergarten<br>d_Busstop<br>d_Attraction<br>d_Parkplaza<br>d_Shopping<br>d_Hospital<br>d_Subway | The distances from each neighborhood centroid to the corresponding nearest PoIs.<br> |
| *Location features* | d_Citycenter<br>d_Huangpuriver | The distances from each neighborhood centroid to the city center and the Huangpu River.<br> |
| *Daily accessed street greenery* | BTA_800<br>BTA_6200<br>Avg_Green<br>Combined_assessment | Local street accessibility for pedestrians (sDNA 800 m)<br>Global street accessibility for commuting behaviors. (sDNA 6200 m)<br>Average human-scale street greenery calculated by the SegNet algorithm, based on the Baidu street view image dataset.<br>Combined assessment of both greenery and accessibility<br> |

## 3. Analysis

### 3.1. Data Preparation

To ensure the reliability of the regression analysis, we conducted preliminary tests to check for multicollinearity before conducting the regression, featuring an examination of the correlation analysis. The correlation test showed that the correlation coefficients between all the variables were less than 0.8, indicating that the degree of multicollinearity in our models might not be sever (Figure 10). Moreover,

a logarithm transformation was performed on several variables according to the average and standard deviation of these variables.

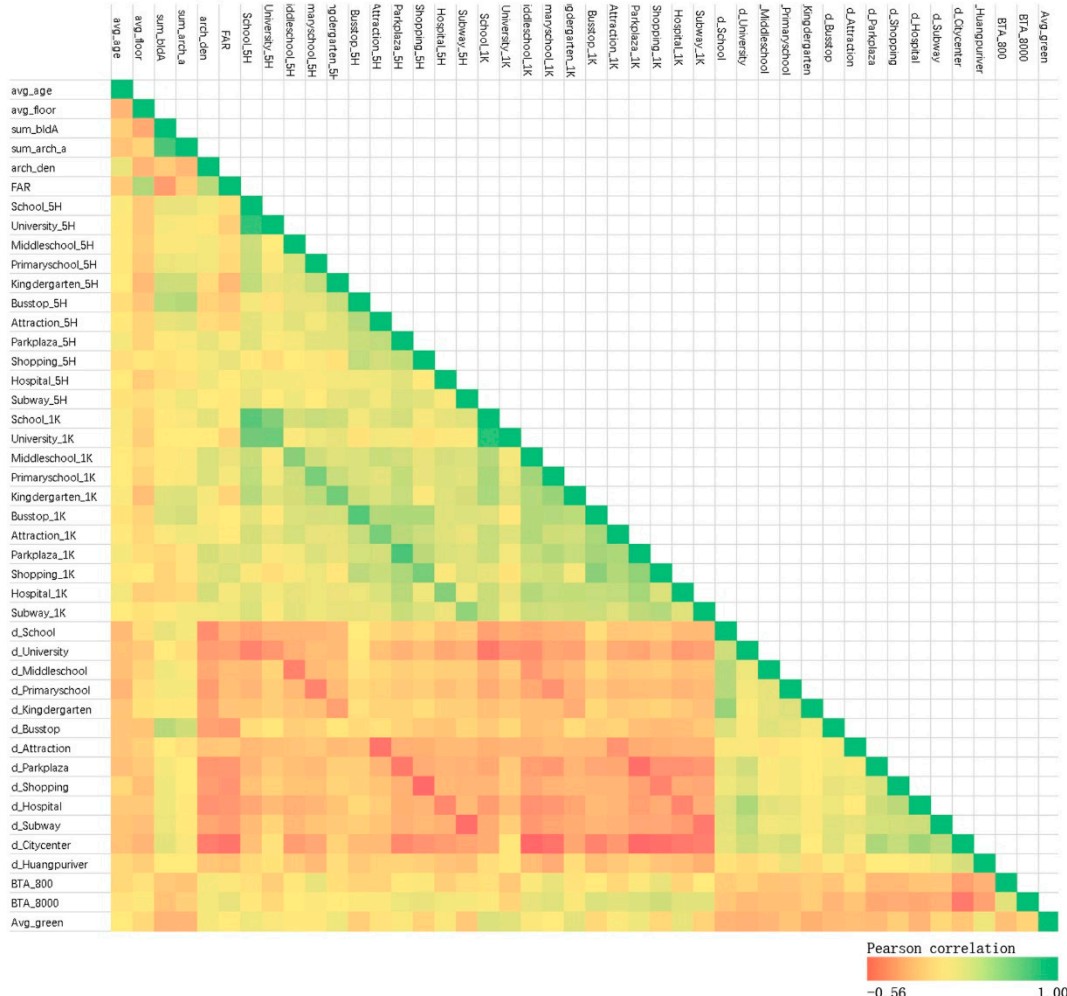

**Figure 10.** A correlation matrix between all dependent and independent variables. Except the situation that two inputs were the same (Pearson correlation coefficient = 1.0), the correlation coefficients between all the rest variables were less than 0.8.

### 3.2. Regression Analyses via the Linear Regression Model

We employed a stepwise linear regression in SPSS to run three models: the initial one without any indicators of daily accessed street greenery (Model 0); the second with visible street greenery and street accessibility as categories (Model 1); and the third with a combined assessment of greenery and accessibility as categories (Model 2). The stepwise method ran multiple regression a number of times while simultaneously removing those that were not important or have problems in variance inflation factor (VIF). By this way, the final model only included significant variables selected in a scientific way.

To avoid multicollinearity, the combined assessment of greenery and accessibility and of visible street greenery and street accessibility were put into different models. As shown in Table 2, adding either visible street greenery and street accessibility or a combined assessment of greenery and accessibility would lead to a significant improvement in the R square and a decrease in the standard error of the estimate. Moreover, a robustness check of these results made through double-log regression showed that adding these variables representing human-scale greenery increased R square as well. Therefore, it is clear that the adding of this human-scale quality on streetscape helped increase the interpretation of regression models on housing price.

**Table 2.** Linear regression analyses of the three models.

| | Adjusted R Square | Std Error of the Estimate | Sig. | N |
|---|---|---|---|---|
| Model 0 | 0.268 | 12793.216 | 0.000 ** | 1395 |
| Model 1 | 0.301 | 12501.679 | 0.000 ** | 1395 |
| Model 2 | 0.285 | 12647.556 | 0.000 ** | 1395 |

** < 0.01

The detailed results of Models 1 and 2 are shown in Table 3. In general, all five main categories, i.e., housing and neighborhood features, density of facilities within a certain radius, distances to the closest facilities, location features, and daily accessed street greenery, have significant variables. The majority of variables from housing and neighborhood features and location features are significant, which is quite reasonable since these two categories usually play the most important role in housing prices. For instance, building age and FAR always have negative effects because buyers prefer new apartments and less crowded environments. A shorter distance from the main landscape feature is also an obvious benefit.

**Table 3.** Detailed regression coefficients in Model 1 and 2.

| Model 1 | | | Model 2 | | |
|---|---|---|---|---|---|
| Variables | Standardized Coefficients (Beta) | Sig. | Variables | Standardized Coefficients (Beta) | Sig. |
| Avg_age | −0.073 | 0.002 ** | Avg_age | −0.068 | 0.005 ** |
| Avg_story | 0.062 | 0.011 * | Avg_story | 0.067 | 0.008** |
| Sum_arch_a | −0.108 | 0.001 ** | Sum_bldA | −0.124 | 0.000 ** |
| Arch_den | −0.153 | 0.000 ** | Arch_den | −0.159 | 0.000 ** |
| Kindergarten_5H | 0.097 | 0.001 ** | Kindergarten_5H | 0.130 | 0.000 ** |
| Parkplaza_5H | −0.164 | 0.000 ** | Parkplaza_5H | −0.159 | 0.000 ** |
| Subway_5H | 0.087 | 0.000 ** | Subway_5H | 0.057 | 0.022 * |
| Primaryschool_1K | −0.108 | 0.000 ** | Primaryschool_1K | −0.112 | 0.000 ** |
| Busstop_1K | 0.157 | 0.001 ** | Attraction_1K | 0.064 | 0.025 * |
| Parkplaza_1K | 0.402 | 0.000 ** | Parkplaza_1K | 0.385 | 0.000 ** |
| Hosptial_1K | 0.112 | 0.000 ** | Hosptial_1K | 0.133 | 0.000 ** |
| d_Huangpuriver | −0.196 | 0.000 ** | d_Huangpuriver | −0.167 | 0.007 ** |
| | | | d_Citycenter | −0.079 | 0.031 * |
| d_Primaryschool | 0.063 | 0.015 * | d_Primaryschool | 0.071 | 0.007 * |
| d_Busstop | 0.076 | 0.005 ** | d_Busstop | 0.078 | 0.004 ** |
| d_Parkplaza | 0.063 | 0.008 ** | d_Parkplaza | 0.079 | 0.002 ** |
| d_Attraction | −0.062 | 0.025 * | d_Attraction | −0.067 | 0.018 * |
| green | 0.191 | 0.000 ** | G_L (low) | −0.114 | 0.000 ** |
| BtA8000 | 0.087 | 0.001 ** | G_L(high) | 0.058 | 0.021 * |

** < 0.01, * < 0.05.

In addition, the variable of human-scale, daily accessed street greenery, also has a positive standardized coefficient that is higher than that of many other variables. The comparison of regression coefficients related to daily accessed street greenery is listed in the histograms. The comparison shows that the visible street greenery (0.191) enjoys significantly higher coefficients than those for hospital (0.112), bus stop (0.157), and kindergarten (0.097) within pedestrian distances (Figure 11a). The coefficient of visible street greenery obtains the third-highest regression coefficient in the whole model. This result clearly indicates that an increase in the proportion of greenery in street scenes can be converted effectively into willingness to pay in the real estate market. When compared with greenery-related variables from a top-down viewpoint, e.g., the density of parks and plazas within 1 km (0.402), the regression coefficient of visible street greenery (0.191) does not show a huge difference. Moreover, the variables of global street accessibility (8 km limit) passed the significance test, indicating that city-scale street accessibility may have a positive impact on housing property values. As for the variables interpreting the combined assessment of visible street greenery and street accessibility

(Figure 11b), low value for the co-presence of local-scale accessibility and eye-level greenery showed negative, significant effect on housing prices, compared with the medium value. In turn, high value for the co-presence of local-scale accessibility and eye-level greenery showed positive, significant effect on housing prices. The results indicate that this human-scale street quality may have a positive impact on housing property values as well. Similar results can be found in the double-log regression analysis working as the robustness check. The visible street greenery obtains significantly higher coefficients than many facilities. The changing of values representing the co-presence of accessibility and eye-level greenery would lead to similar and significant impacts as well.

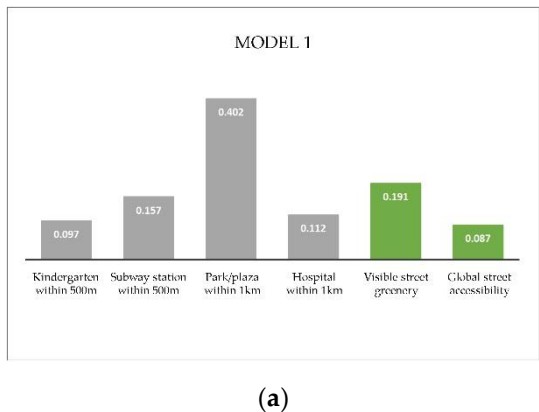 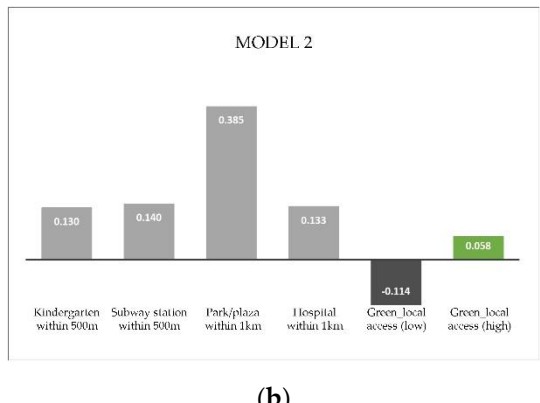

|    (**a**)    |    (**b**)    |

**Figure 11.** Comparing the standardized coefficients between daily accessed street greenery with other categories: (**a**) standardized coefficients for several variables in Model 1; and (**b**) standardized coefficients for several variables in Model 2.

In fact, considering the density of facilities and proximity to parks/plazas are largely determined by the specific site location and existing land-use plans, there is little chance for transformation or intervention. Therefore, the economic benefit of daily accessed street greenery is rather important since it provides a more flexible and feasible way of promoting housing property values in a developed urban context.

We also checked the negative results for density of primary school (Primaryschool_1K) and urban parks (Parkplaza_5H), which appeared unusual. Theoretically, the increase in important facilities should lead to an increase in buyers' willing to pay. Nevertheless, many neighborhoods in Shanghai featuring good housing facilities within a close distance are old socialism workers' neighborhoods built between the 1950s and 1970s. The poor condition of the housing and neighborhood features plays a more important role in housing price and thus leads to the abnormal regression coefficients mentioned above.

## 4. Discussion

### 4.1. The Positive Impact of Daily Accessed Street Greenery on Housing Price

Based on the regression analyses above, we identified the positive impact of indicators from daily accessed street greenery on housing price. The adding of either visible street greenery and street accessibility, or a combined assessment of human-scale greenery and accessibility would improve the interpretation of the current hedonic price model. These findings confirm the economic performance of the human-scale streetscape. Specifically, the standardized coefficient of visible street greenery is even higher than the corresponding coefficients of many facilities' densities. There is also no great difference between the regression coefficient between visible greenery and the density of parks and plazas within pedestrian distance. In addition, the co-presence of street accessibility at local scale and visible greenery is also significant for housing prices.

Some examples are presented as a direction illustration of the economic performance of daily accessed street greenery (Figure 12).

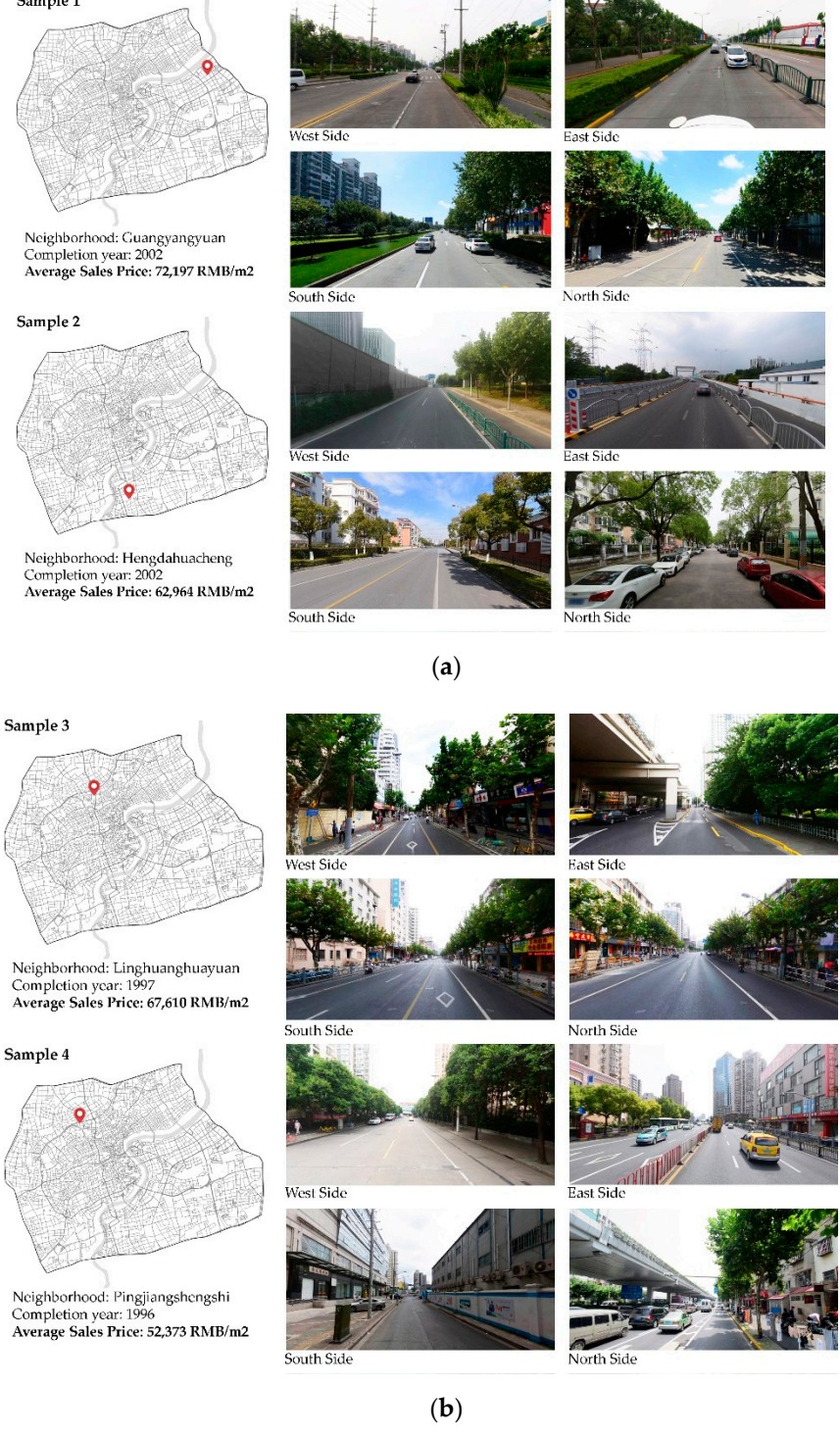

**Figure 12.** Examples illustrating the economic performance of daily accessed street greenery: (**a**) Samples 1 and 2 located in Pudong new district; and (**b**) Samples 3 and 4 located in the inner ring road area.

As shown in the Figure 12a, two neighborhoods built at the same time (2002) were selected as the samples. Both are located in Pudong new district that was quickly built up in recent decades. Meanwhile, Samples 1 and 2 obtain similar sizes containing 20 buildings, location features, density

and distance of facilities. The streetscapes around these two sample sites were recorded as well. Considering these similarities between the two samples, the different co-presences of visible street greenery and accessibility, i.e., from high to medium value, might play an important role in this observed price variance of 72,197 RMB/m$^2$ vs. 62,964 RMB/m$^2$. A similar trend can be found in Figure 12b. Both Samples 3 and 4 are in the inner ring road of Puxi area, the old part of Shanghai. These two samples also obtain a series of similar features but have different housing prices. The decreasing of visible street greenery from high to medium value might lead to this observed price variance of 67,610 RMB/m$^2$ vs. 52,373 RMB/m$^2$.

Therefore, we may conclude that, among the implicit attributes, human-scale daily accessed greenery is of great significance for hedonic price models, such that improvements in it can be supposed to support willingness to pay in the real estate market more effectively. This quantitative measurement provides us with strong support for the significance of human-scale street greenery, making it an important issue in urban greening policy for urban planners and decision makers, as well as a reference for housing price predictions.

*4.2. The Influential Mechanism between Daily Accessed Street Greenery and Housing Price*

Another important issue that needs to be clarified is the mechanism of influence between daily accessed street greenery and housing price. In other words, does high daily accessed street greenery really increase people's willingness to pay? We need to discuss another possibility first, i.e., this significant effect between daily accessed street greenery and housing price might be caused by an opposite approach. It might be the high housing price that contributes to high property taxes, and then these taxes as government revenue would allow local municipality to build and afford high-quality greenery on the streets.

Further study in this direction found this potential approach can be excluded. In contrast to the United States where property taxes are collected by the local municipality, the property tax in China is collected by the central government. Therefore, the local municipality would not obtain any direct income from high property prices, and thus this potential approach we discussed above does not exist. Based on that, we can finally conclude that daily accessed street greenery, as representative of a human-scale streetscape, improves people's perceptual feelings about the property and finally leads to a positive impact on housing prices.

**5. Conclusions**

*5.1. Concluding Remarks*

This study attempted to measure the economic performance of daily accessed street greenery in an era of new urban science. By combining new urban data, including many street view images, PoIs, and detailed street networks from OSM, with new tools, including machine-learning algorithms and space syntax, the level of accessed visible street greenery in residents' daily lives was identified and its impact analyzed in relation to housing prices. Through the standardized coefficients obtained from the linear regression model, we compared the influential weights of daily accessed street greenery with other traditional indicators in hedonic price models. This study provides scientific and quantitative support for the significance of this kind of human-scale street greenery, making it an important issue in urban greening policy for urban planners, developers, and decision makers.

*5.2. Policy Implication*

In light of the finding above, the objectively measured economic values and potential of human-scale greenery may encourage developers and decision makers to pay more attention to human-scale street-level greenery in the process of place making. For real estate developers, paying extra attention to high-quality street greenery for both the internal and surrounding streets of

their projects may be more profitable than previously believed. As important feature of a location, daily accessed street greenery is able to bring good prospects of gain.

In addition, the results can act as a reference for the government in setting up associated indicators or standards for encouraging human-scale street-level greenery, thereby realizing the refined management of urban space. Higher human-scale qualities may attract high-net-worth families, which might then benefit an area. Meanwhile, by modifying the existing related policies, the government may also require the real estate developers to take on more responsibilities in the improvement of public realms.

Moreover, the relatively high impact of visible street greenery in Shanghai implicates the current spatial layout of urban parks might be inappropriate as it missed the concern of human-scale perceptions. The decision makers may take the urban context into consideration in the future planning of streetscapes and street layouts to exploit the value of human-scale street-level greenery to the full. In this regard, it is important to incorporate the existing large-scale parks by linking them to street-level green corridors, thus creating a more complete and multi-level system of urban green infrastructure. This understanding may lead to a review of current urban greenery policy replying on satellite images with a top-down viewpoint. The adding of human-scale viewpoint would be necessary.

### 5.3. Limitations and Next Steps

Despite the contribution of this study in measuring the economic performance of daily accessed street greenery, it had several limitations that require further exploration. Although this study mainly focused on identifying the relative impacts of human-scale indicators rather than on building a good regression model of housing prices, its R square still has some scope for improvement.

This situation might be caused by three reasons. First, there was a sharp rise of Shanghai housing price in 2016, which led to around 50–60% increase in only nine months [55]. This irrational jumping of housing price highlights the financial and speculative attributes of housing property, which tends to push buyers to overlook built environment factors that were important as well. The effect of housing market sentiment has been recognized by empirical studies in China [56]. This kind of housing market sentiment might limit the accuracy of classical hedonic price model in Chinese megacities. Second, the typical urban morphology in Shanghai is high-rise, collective residential building blocks containing many buildings and private inner streets. Therefore, it is difficult to use a mean property price to represent a whole community considering the inherent difference among hundreds of apartments. Using buildings rather than street blocks as analytical units would lead to a higher R square, which can be found in existed studies [57]. Nevertheless, it would be hard to integrate urban streetscape features with inner buildings. Perhaps running the analysis at the scale of street blocks would be the best choice to achieve the co-presence of urban streetscape features with housing features. Considering the main focus was studying the effects of human-scale streetscape features rather than on achieving a good regression model with high R square, the current result using street blocks could be accepted. Moreover, some zoning settings, e.g., school district, play an important role on housing price at Shanghai. A good school district usually brings a high premium for housing price, according to empirical studies in other Chinese cities [58]. Nevertheless, it is hard to include these settings into current analyses considering the lack of digitalized and geo-referenced data.

Therefore, it is important to handle these issues well in the sequent studies. On the one hand, a continuing data collection over years on both housing price and built environment factors would be made in our following studies for selecting an appropriate period that is able to control the effect from housing market sentiment. On the other hand, endeavors would be made to include important zoning settings, such as school district, into our model. Manually-based digitization of these settings would be made in our following studies. Besides these improvements on data input, we will attempt to add in hierarchical linear model (HLM) into analyses, which might help to consider the hierarchical structure of the massive input data and improve the goodness of fit [27]. Final improvements in this analytical model might provide data of greater specificity for the government and policy makers.

**Author Contributions:** Conceptualization, Y.Y. and J.F.; methodology, Y.Y.; formal analysis, Y.Y.; investigation, J.F.; resources, H.J. and H.X.; data curation, H.J.; writing—original draft preparation, H.X. and Y.Y.; writing—review and editing, Y.Y., J.F. and H.J.; supervision, Y.Y., D.W. and J.F.; and funding acquisition, Y.Y., D.W. and J.F.

**Funding:** This research was funded by Natural Science Foundation of China, grant numbers 51708410, 51608368 51838002, and 41771170.

**Acknowledgments:** We want to thank the editor and anonymous referees for their kind help. We also thank the assistance from Haoming Tang in data preparation.

**Conflicts of Interest:** The authors declare no conflict of interest.

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
