# Peer review of "Daily Accessed Street Greenery and Housing Price: Measuring Economic Performance of Human-Scale Streetscapes via New Urban Data"

_sustainability, doi:10.3390/su11061741_

Reviewer 1 Report

The conclusions are very poor. I suggest to extend the discussions and devote a final section to the policy implications.         

Author Response

Response to Reviewer 1 Comments

Point 1: The conclusions are very poor. I suggest to extend the discussions and devote a final section to the policy implications.

Response 1: Many thanks for this detailed suggestion. We have re-organized the discussion and conclusion sections following the reviewer’s comment (Page 17-19, Line 462-531). The conclusion section now has three sub-sections: “concluding remarks” given a short conclusion, “policy implications” given a discussion the potential implication of findings from this study, and “limitations and next steps” claiming the current shortcomings and our plans in the next step. Meanwhile, the discussion on policy implications has been expanded as well.

We highly appreciate the reviewer taking the time to offer us comments and insights related to the paper. We would be glad to respond to any further questions and comments that you may have.

Reviewer 2 Report

The paper analyses the potential economic effect on real estate prices of daily accessed street greenery on a popular city: Shangai.

The paper benefits from several positive attributes, and then deserves publication in my opinion. I have enjoyed by reading the different sections, and how the authors have introduced some topics, and the different techniques and technologies they have used to conduct the research.

There are some very minor points that authors could addressed –if appropriate-:

1. The Introduction section is not large enough to comprehend the topic in deep. I would suggest to join this section with the following one: Literature review. In fact, the latter could benefit from including more references related with the topic. For example, the subject of real estate pricing in neighborhoods and how variables relate each other depending on the neighborhood considered has been covered in Arribas, I., García, F., Guijarro, F., Oliver, J., & Tamošiūnienė, R. (2016). Mass appraisal of residential real estate using multilevel modelling. International Journal of Strategic Property Management, 20(1), 77-87. Further references could give insight about all dimensions affecting real estate prices in big cities.

2. Figure 3. Which is the unit monetary measure for prices? Is price measured for the whole real estate element, or is considered as price per square meter? We can see some neighborhoods with none, few or a lot of points. Which is the minimum quantity of points considered per neighborhood?

3. Line 210. What’s the meaning of PoI? I guess the acronym stands for “Point of Interest”, but authors should clarify this point in the paper.

4. Line 231. Authors consider street segments. That way, four pictures are considered for all street segments. However, we can see that some streets are very large and some others can be very short. This translates in streets with much green areas in the middle of the street, not just in the corners, which makes difficult to measure how green they are, according to the method used by the authors. I think this point could be considered as a limitation of the research.

5. Table 1. Despite authors include all figures in separate files, and the quality of these figures is really good, figures included in tables are poor. This can make difficult to read the paper. I would suggest to enlarge the pictures to improve the quality of the draft.

6. Regression. I think tables in the main text and appendices give enough information about the regression models and the results obtained. However, I would suggest the authors to include a stepwise regression. This way, the final model would include only significant variables and these variables would have been selected in a scientific way.

There is only an important drawback: the low values registered in the R2. In the literature, authors could find a lot of references with determination coefficients close to 0.8 or 0.9. Why the reported R2 is so low? It is a bit frustrating to see that only 30% of variability in prices can be explained by variables selected by authors. A good discussion regarding this point can be considered.

7. I would also suggest to include the correlation matrix between dependent and independent variables. I realize that the dimension of this matrix can be too large, but authors could present their results as a figure instead of a matrix with correlations coefficients.

Finally, I want to congratulate the authors for the high quality of the paper and encourage them to continue in the same line. Good luck in the future.

Author Response

Response to Reviewer 2 Comments

The paper analyses the potential economic effect on real estate prices of daily accessed street greenery on a popular city: Shanghai.

The paper benefits from several positive attributes, and then deserves publication in my opinion. I have enjoyed by reading the different sections, and how the authors have introduced some topics, and the different techniques and technologies they have used to conduct the research.

There are some very minor points that authors could addressed –if appropriate-:

Point 1: The Introduction section is not large enough to comprehend the topic in deep. I would suggest to join this section with the following one: Literature review. In fact, the latter could benefit from including more references related with the topic. For example, the subject of real estate pricing in neighborhoods and how variables relate each other depending on the neighborhood considered has been covered in Arribas, I., García, F., Guijarro, F., Oliver, J., & Tamošiūnienė, R. (2016). Mass appraisal of residential real estate using multilevel modelling. International Journal of Strategic Property Management, 20(1), 77-87. Further references could give insight about all dimensions affecting real estate prices in big cities.

Response 1: Thanks a lot for your kind help. We have revised the previous sections of Introduction and Literature Review following the reviewer’s suggestion. Now these two sections have been reorganized as the new Introduction section (Page 1-4, Line 34-150). Many references related with the topic have been added, which included the one mentioned by the reviewer (Arribas et al., 2016) and some other recent papers as well. We hope now the Introduction section is large enough to comprehend the topic in deep.

Point 2: Figure 3. Which is the unit monetary measure for prices? Is price measured for the whole real estate element, or is considered as price per square meter? We can see some neighborhoods with none, few or a lot of points. Which is the minimum quantity of points considered per neighborhood?

Response 2: Many thanks for your detailed suggestion. Revisions have been made in both Figure 3 and related text. The unit monetary used herein is Chinese currency (RMB) per square meter, which has been claimed in both Figure 3 and section 2.3.1 (Page 5, Line 199).  Some blocks without points means a lack of data on housing price. These blocks are marked as grey color, which usually obtain other urban functions, like commercial facilities or green parks. Most street blocks contain one point. Some blocks contain a few of points due to two reasons: 1) some residential communities were built in phases and thus contains several records on the website, 2) several street blocks contained more than one parcels. In these communities contained more than one record, the final record of building age and property price were averaged among these collecting points. These explanations have been added in the main text as well (Page 5, Line 199-207).

Point 3: Line 210. What’s the meaning of PoI? I guess the acronym stands for “Point of Interest”, but authors should clarify this point in the paper.

Response 3: We apologize for this mistake. Yes, it stands for “points-of- interest”. A revision has been made on current line 231.

Point 4: Line 231. Authors consider street segments. That way, four pictures are considered for all street segments. However, we can see that some streets are very large and some others can be very short. This translates in streets with much green areas in the middle of the street, not just in the corners, which makes difficult to measure how green they are, according to the method used by the authors. I think this point could be considered as a limitation of the research.

Response 4: We apologize that we did not explain it clearly in our previous manuscript. Actually, the collection of street view images is based on sample sites rather than street segments. Each sample site would collect four pictures. And the distance between each sample site along streets is a fixed number (around 40 meters). This short distance would help to catch the changing of street greenery. Therefore, the length of street segment might not affect the result.

The detailed revision is attached below (Page 7, Line 252-255):

The 69,137 sample sites were generated along streets from the OSM street network with a total length of 2,611,079 meters, in the city center of Shanghai. Each sample site contained four SVIs with the size of 480 x 360 pixels were enough to achieve a panoramic view of the surrounding environment (Figure 5).

After the calculation of visible greenery on street view images and sample sites, we were then able to calculate the value along a single street segment as an overall greenery index (Page 8 Line 282-284).

Point 5: Table 1. Despite authors include all figures in separate files, and the quality of these figures is really good, figures included in tables are poor. This can make difficult to read the paper. I would suggest to enlarge the pictures to improve the quality of the draft.

Response 5: Thanks for your kind help on the arrangement of Table 1. We agree that the pictures inside the Table 1 are too small to be read. Revisions have been made to enlarge the pictures (Page 10-12).

Point 6: Regression. I think tables in the main text and appendices give enough information about the regression models and the results obtained. However, I would suggest the authors to include a stepwise regression. This way, the final model would include only significant variables and these variables would have been selected in a scientific way.

There is only an important drawback: the low values registered in the R2. In the literature, authors could find a lot of references with determination coefficients close to 0.8 or 0.9. Why the reported R2 is so low? It is a bit frustrating to see that only 30% of variability in prices can be explained by variables selected by authors. A good discussion regarding this point can be considered.

Response 6: Thanks a lot for your help. We have used the stepwise method to re-run the regression. Related text and tables in the section 3.2 have been revised (Page 13-15, Line 346-403). Now the final model only shows significant variables selected in a scientific way.

We also agree that the low value of R2 is a shortcoming. A detailed discussion regarding this point has been added in section 5.3 “limitations and next steps” (Page 17, Line 496-531). Three main reasons caused this low R2 and our following studies have been discussed.

Point 7: I would also suggest to include the correlation matrix between dependent and independent variables. I realize that the dimension of this matrix can be too large, but authors could present their results as a figure instead of a matrix with correlations coefficients.

Finally, I want to congratulate the authors for the high quality of the paper and encourage them to continue in the same line. Good luck in the future.

Response 7: Many thanks for your kind suggestion and warm encouragement. A correlation matrix between dependent and independent variables has been added following the reviewer’s suggestion (Figure 10 at Page 13).

We highly appreciate the reviewer taking the time to offer us comments and insights related to the paper. We would be glad to respond to any further questions and comments that you may have.

Reviewer 3 Report

In general I think this is a very good paper and I am especially impressed by the data collection methods. Just a few notes:

- Maybe some checks on the robustness of the results could have been included, eg. testing different functional forms.

- It would be interesting with some examples where the size of the difference in value for similar houses/apartments located close to and further away from green space.

- The discussion at the top of p 25 about property taxes are confusing. If the property tax is determined in the same way for all properties it cannot affect the link between green space and price. This section should either be deleted or expanded with more institutional detail. The crucial issue is the one about robustness mentioned above.

- The section 5.4 should perhaps be moved to the conclusion which could be expanded with more of the results..

Author Response

Response to Reviewer 3 Comments

In general I think this is a very good paper and I am especially impressed by the data collection methods. Just a few notes:

Point 1: Maybe some checks on the robustness of the results could have been included, eg. testing different functional forms.

Response 1: Thanks for your kind suggestion. We have added a robustness check through the double-log regression. It shows that the adding of these variables representing human-scale greenery would increase R squares among the models as well (Page 14, Line 357-359).  Moreover, the visible street greenery obtains significantly higher coefficients than many facilities. The changing of values representing the co-presence of accessibility and eye-level greenery would lead to similar and significant impacts as well (Page 15, Line 393-396).

Point 2: It would be interesting with some examples where the size of the difference in value for similar houses/apartments located close to and further away from green space.

Response 2: A new Figure 12 and related text have been added in the discussion section following the reviewer’s suggestion (Page 16-17, Line 422-440).

Point 3: The discussion at the top of p 25 about property taxes are confusing. If the property tax is determined in the same way for all properties it cannot affect the link between green space and price. This section should either be deleted or expanded with more institutional detail. The crucial issue is the one about robustness mentioned above.

Response 3: Many thanks for your detailed suggestion. A revision has been made in this part to add in more detailed explanations. We prefer to keep it because it might be necessary to clarify the mechanism of influence between daily accessed street greenery and housing price. In other words, does high daily accessed street greenery really increase people’s willingness to pay? An opposite approach should be discussed first. It might be the high housing price that contributes to high property taxes, and then these taxes as government revenue would allow local municipality to build and afford high-quality greenery on the streets.

Further study in this direction found this potential approach can be excluded. In contrast to the United States where property taxes are collected by the local municipality, the property tax in China is collected by the central government. Therefore, the local municipality would not obtain any direct income from high property prices, and thus this potential approach we discussed above does not exist. Based on that, we can finally conclude that it is the daily accessed street greenery, as representative of a human-scale streetscape, that improves people’s perceptual feelings about the property and finally leads to a positive impact on housing prices. Detailed revisions are placed at Page 17 (Line 447-461).

Hope it is clear now. We can also delete this part in next round of revision if the reviewer feel it is still confusing.

Point 4: The section 5.4 should perhaps be moved to the conclusion which could be expanded with more of the results.

Response 4: Many thanks for this detailed suggestion. We agree that the conclusion section was weak. Now we have re-organized it following the reviewer’s comment (Page 17-18, Line 462-531). The conclusion section now has three sub-sections: “concluding remarks” given a short conclusion, “policy implications” given a discussion the potential implication of findings from this study, and “limitations and next steps” claiming the current shortcomings and our plans in next step.

We highly appreciate the reviewer taking the time to offer us comments and insights related to the paper. We would be glad to respond to any further questions and comments that you may have.
